# High Prevalence of Syphilis and Syphilis/HIV Coinfection among Men Who Have Sex with Men Who Attend Meeting Places in Mexico

**DOI:** 10.3390/pathogens12030356

**Published:** 2023-02-21

**Authors:** Dayana Nicté Vergara-Ortega, Andrés Tapia-Maltos, Antonia Herrera-Ortíz, Santa García-Cisneros, María Olamendi-Portugal, Miguel Ángel Sánchez-Alemán

**Affiliations:** 1Centro de Investigación Sobre Enfermedades Infecciosas, Instituto Nacional de Salud Pública, Cuernavaca 62100, Mexico; 2Plan de Estudios Combinados en Medicina (PECEM), Facultad de Medicina, Universidad Nacional Autónoma de México, Ciudad de México 04510, Mexico

**Keywords:** men who have sex with men, syphilis, HIV, syphilis/HIV coinfection, meeting places

## Abstract

Men who have sex with men (MSM) are disproportionately affected by syphilis, HIV, and syphilis/HIV coinfection. Antiretroviral therapy (ART) prevents HIV transmission but does not impede the spread or acquisition of syphilis. Information about syphilis/HIV coinfection among MSM is scarce. We aimed to determine the prevalence of syphilis/HIV coinfection in a national sample of MSM who attend meeting places (such as movies, clubs, gay bars, saunas, shopping malls, and others referred to by the same MSM participants of the study) in Mexico to evaluate factors associated with syphilis, and to compare the prevalence rates of syphilis between the current survey and DGE data. We performed a laboratory diagnosis to determine the rates of syphilis and HIV among the included MSM. The national and regional prevalence of syphilis was calculated. HIV and coinfection prevalence were determined only for the survey. All prevalence rates included _95%_CIs. Descriptive, bivariate, and multivariate analyses were performed. The national prevalence rates of syphilis, HIV, and coinfection were 15.2%, 10.2%, and 5.7%, respectively. The region with the highest prevalence rate was Mexico City (39.4%). The center region, minimal “goods” (i.e., a minimal number of material possessions such as a car or dryer, etc., which served as a proxy for low economic income level), use of “inhalant drugs”, “HIV infection”, “sexual intercourse” only with men, “rewarded sex”, and “youngest age at first sexual encounter or debut” were risk factors for syphilis. In general, regional prevalence of syphilis was higher in the survey (2013) and DGE data from 2019 than in the DGE data from 2013. Similar to other countries, Mexico needs to assess elements around not only syphilis and HIV infections but also syphilis/HIV coinfection, and preventive measures focusing on MSM are needed.

## 1. Introduction

At present, syphilis is a re-emerging infection. Men who have sex with men (MSM) are disproportionately affected by sexually transmitted infections (STIs), including syphilis, HIV, and syphilis/HIV coinfection [1,2,3]. Various factors that have influenced the re-emergence of syphilis, including risk compensation, serosorting, seroposition, and an increase in the prevalence of condomless sex [1,3,4]. Risky sexual behaviors remain a matter of concern, given that up to 25% of seropositive MSM are unaware of their HIV status [1]. Additionally, while antiretroviral therapy (ART) prevents HIV transmission by making the viral load undetectable, it does not impede the spread and acquisition of other STIs, including syphilis, especially during unprotected sex [1,5]. This could explain why the incidence of syphilis among MSM increases at a faster rate than that of HIV in some countries [3,6].

If an HIV-positive person becomes infected with syphilis, the acquired bacterial infection can cause a transitory reduction or even a permanent loss of CD4^+^ T cells [4,5] and can increase the viral load [3,5]. The genital ulcers caused by syphilis may stimulate the recruitment of antigen-presenting cells and CD4^+^ T cells at the lesion site; spirochetes also induce the expression of CCR5 in macrophages [1,7]. These immunological consequences of syphilis may thus enhance the risk of acquiring HIV during sexual intercourse [4]. On the other hand, HIV infection can have effects on the course of syphilis, and it can also have a faster and more aggressive presentation, such as concurrent primary and secondary syphilis. People living with HIV (PLWH) can develop deeper and numerous syphilitic ulcers, which in turn makes an individual more likely to transmit or acquire infections [1,3,5,7]. Furthermore, the rate of neurosyphilis in earlier stages is higher in PLWH; this difference is linked to an inability to control the progression of the infection, leading to a rapid progression to neurosyphilis [2,7].

In Mexico, information regarding the epidemiology of syphilis/HIV coinfection in MSM is scarce. The objective of this study was to determine the prevalence of syphilis/HIV coinfection in a national sample of MSM who attend meeting places, to evaluate the factors associated with syphilis infection, and to compare the data from the current survey with public data about MSM with syphilis in Mexico.

## 2. Materials and Methods

### 2.1. Survey Design and Fieldwork

A national survey of MSM who attend meeting places was performed during 2013 in 24 states across 6 geographic regions: Mexico City, South, Center, Northeast, Northwest, and West. Therefore, this survey is nationally and regionally representative [8,9]. In this context, meeting places are sites at which the MSM population converge and spend their free time, such as movies, clubs, gay bars, shopping malls, saunas, and others. All the methods in the study were performed under the guidelines, regulations, and instructions of ethics and biosafety Instituto Nacional de Salud Pública (INSP) committees. After the complete observation and evaluation of the study, we obtained approval number 1393. Prior to informed consent, each MSM participant answered a self-report questionnaire, and a dried blood spot (DBS) sample was collected. The study population for our study was 4977 MSM surveyed in 2013 [9,10].

### 2.2. Laboratory Test

Detection of HIV infection was performed with a chemiluminescent microparticle immunoassay with DBS samples using the HIV Ag/Ab Combo, Architect, Abbott Diagnostics Division™ (sensitivity to identify antibodies against HIV-1/2 100%, p24 antigen analytical sensitivity <50 pg/mL, and specificity ≥99.5%), following the manufacturer’s instructions [11,12]. Detection of antibodies against *T. pallidum* was carried out with the ELISA kit SYPHILIS Biokit 3.0^®^ (Werfen OEM Group, Barcelona, SPAIN) (sensitivity 97.4% and specificity 100%) [13], previously standardized to use DBS. Elution was performed with 150 μL of buffer and an incubation period of 1 h at 25 °C; an index was calculated with the calibrator. The negative samples were those with an index of 0.8 or less, the positive had an index of 2.0 or greater and the indeterminate had an index between 0.8–2.0. The latter were confirmed with an agglutination test (Serodia^®^TP-PA, Fujirebio Inc., Tokio, JAPAN) [14].

### 2.3. Statistical Analyses

A descriptive analysis of sociodemographic and sexual behavior variables was carried out. Subsequently, bivariate and multivariate logistic regression analyses were performed to determine the factors associated with *T. pallidum* infection and syphilis-HIV coinfection and the odds ratios (ORs). All analyses were carried out with confidence intervals at 95% (_95%_CI) and *p* < 0.05 using SPSS software version 23.0. As a limitation, we did not conduct any statistical method of correction in multiple comparisons.

### 2.4. Public Data Analyses

We collected data on syphilis infections reported in Mexico from the web application of Dirección General de Epidemiología (DGE), Secretaría de Salud of Mexico (Available online: http://www.dgis.salud.gob.mx/contenidos/basesdedatos/BD_Cubos_gobmx.html (accessed on 28 October 2022) [15]. In this database, each state details a raw number of infections and the prevalence by population type and year. For this study, we used the data about MSM to estimate the prevalence and calculate the _95%_CI of syphilis by region, considering the same regions and states of the survey.

## 3. Results

### 3.1. Descriptive Analysis

In the study population, Mexico City was the “region” with most individuals (27.9%), and the region with the fewest individuals was the South (6.0%). Nearly 70% of MSM surveyed were under the age of 29. As a proxy variable of economic income level, the number of “goods” of individuals (i.e., a categorized number of material possessions such as a car, washing machine, dryer, or television, etc.) were concentrated in the medium and high categories. We found that 61% percent of the surveyed population had jobs and 26.0% were students. Only 235 individuals (4.7%) reported using “inhalant drugs” in the last year (12 months). “HIV infection” was confirmed in 15.9% of MSM, and more than 50% of the individuals never underwent an “HIV test”. In the last year, 88.0% of individuals had “sexual intercourse” only with men and 26.7% had exchanged sex for money or gifts (“rewarded sex”). Almost 44.0% of the population had one “male sexual partner” in the last month and 5.9% had more than six partners. Of MSM individuals, 32.3% had a “sexual debut with men” when they were younger than 15 years of age (Table 1).

### 3.2. Factors Associated with Syphilis

The estimated national prevalence of syphilis in 2013 was 15.2% (_95%_CI 14.2–16.2). The prevalence of HIV was 10.2% (_95%_CI 9.3–11.0) (Table 1). The prevalence of syphilis ranged from 15.1% (_95%_CI 12.9–17.6) in the West to 24.7% (_95%_CI 22.0–27.5) in the center (Table 2).

Regarding the “geographic regions” analyzed, individuals in the center had almost twice the risk of being positive for syphilis compared to the other regions (1.9; _95%_CI 1.5–2.5). The risk of being positive for syphilis increased with age due to the increase in exposure. Individuals who reporting having a low quantity of “goods” presented the highest risk for being positive for syphilis (1.3; _95%_CI 1.1–1.7). Regarding “occupation”, economic income was found to be associated with the risk of becoming positive for syphilis because the categories of employee (1.4; _95%_CI 1.1–1.7), own business (1.6; _95%_CI 1.2–2.3), and other occupations (1.9; _95%_CI 1.0–3.4) resulted in a higher risk of syphilis than unemployed MSM and MSM who were students. The use of “inhalant drugs” in the last year was associated with a higher risk of syphilis among individuals who consumed them at any time (1.5; _95%_CI 1.1–2.1) than among those who never did. People with laboratory-confirmed “HIV infection” have a two-fold risk of being positive for syphilis compared with individuals without HIV (2.0; _95%_CI 1.7–2.4). Individuals who reported having had a “test for HIV” in the last year had a higher risk of syphilis than those who did not have a test for HIV (1.3; _95%_CI 1.1–1.5) (Table 2).

Regarding the variables of sexual behavior, the results show that MSM who have “sex only with men” (in the last 12 months) have a higher risk of presenting with syphilis (1.4; _95%_CI 1.1–1.8) than those who have sex with men and women. The results are similar with respect to “rewarded sex”, such that those who reported a history of “rewarded sex” have a higher risk of syphilis (1.3; _95%_CI 1.1–1.5). We observed a higher risk of syphilis among individuals with six or more “male sexual partners in the last month” (1.4; _95%_CI 1.0–2.0) than among those who responded “neither” to this item. Finally, MSM who reported having their “sexual debut with men” at an early age (8–10 years) had a higher probability of having syphilis (2.0; _95%_CI 1.2–3.3) than those who had their first such encounter at an older age (Table 2).

### 3.3. Syphilis/HIV Coinfection

The prevalence of syphilis/HIV coinfection was estimated to be 5.7% (_95%_CI 5.1–6.4). Table 3 shows the following columns: syphilis- and HIV-negative (double negative), syphilis-positive and HIV-negative (syphilis prevalence), syphilis-negative and HIV-positive (HIV prevalence), and syphilis- and HIV-positive (coinfections).

Based on the data, an increasing trend was observed in Mexico City from double negatives (syphilis- and HIV-negatives) to coinfected people (25.9% vs. 39.4%). The opposite effect was observed in the Northwest (14.3% vs. 12.3%). An increase in percentage in the columns is observed for “age”, specifically among those aged 30–34 years old and those aged 35 years old or older (12.0% vs. 20.1% and 14.9% vs. 28.9%, respectively), such that the lowest percentage was observed for double negatives and the highest percentage was observed for coinfections. In contrast, the youngest individuals (under 24 years old) show the opposite trends, such that the prevalence of double negatives was higher than that of coinfection. Of all the data, MSM ≥35 years presented the highest prevalence of coinfection (28.9%).

MSM who reported having minimal “goods” (0–3) presented lower rates of coinfection than MSM with other statuses (14.8% vs. 44.7% and 40.5%). The rate of coinfection among employed people was 66.2%, while the rate among students was only 7.6%. Similarly, only 7.7% of “inhalant drug” users reported coinfection. Syphilis/HIV coinfection was more common among people who had an “HIV test” in the last year (59.5%), in MSM who have “sexual intercourse only with men” (94.7%), and in those who do not practice “rewarded sex” (60.9%). Regarding the number of “male sexual partners”, the highest prevalence of coinfection was observed among those with one partner (35.9%). Additionally, regarding the “age at sexual debut”, we observed an increase in coinfection rates among the “8–10 years old” and “11–15 years old” groups (double negative rate of 2.9% and 27.3%, respectively; coinfection rates of 8.5% and 34.2%, respectively). Among those who responded 16–20, 21–25, and ≥26 years old, for the variable “sexual debut”, there was a decrease in the prevalence when comparing double negative (higher) status to coinfection (lower) rates.

### 3.4. Comparison of Syphilis Prevalence among MSM: Public Data vs. National Survey in Meeting Places

Data collected from DGE regarding the number of syphilis cases among MSM in Mexico was the basis for the prevalence estimate and _95%_CI in each geographic region. The states included herein were the same as those considered in the national survey of MSM who attend meeting places: Northwest: *Baja California, Chihuahua, and Sonora*; Northeast: *Nuevo León, San Luis Potosí, and Tamaulipas*; West: *Jalisco, Guanajuato, and Aguascalientes*; Center: *Puebla, Tlaxcala, Morelos, Oaxaca, Veracruz, Guerrero, and Hidalgo*; Mexico City; and South: *Yucatán, Quintana Roo and Campeche* (Figure 1).

The analysis of the regional prevalence in 2013 with DGE data showed five regions under 10.0%, except for center (14.0%; _95%_CI 12.1–16.1). The first comparison was between the prevalence rates reported in this database and the regional prevalence obtained in the survey from 2013. All regions reported at least a two-fold increase in their prevalence (except center, which reported a significant increase from 14.0% vs. 24.7%) a 20 percent increase or greater, as in Mexico City (4.5%; _95%_CI 2.1–9.3 vs. 24.0%; _95%_CI 21.8–26.3). Another comparison was performed with data from DGE 2013 and data from DGE 2019. In this case, despite the increase in prevalence, the difference was not so high. The highest rates were observed in the South (6.1%; _95%_CI 4.2–8.8 vs. 11.6%; _95%_CI 10.4–13.0). The center was the only region with a decrease in syphilis prevalence: 14.0% (_95%_CI 12.1–16.1) in 2013 and 10.4% (_95%_CI 9.6–11.4) in 2019. Finally, due to the re-emergence of syphilis and the lack of surveys of MSM who attend meeting points in 2019, we hypothesize that there is a high prevalence of syphilis across all regions, but further investigation is needed to confirm this.

## 4. Discussion

This study was performed due to the re-emergence of syphilis in multiple countries, including Mexico, and due to the fact that *T. pallidum* (causal agent) continues to be transmitted among MSM [9,10,16,17].

Recent data released by the World Health Organization (WHO) have shown that the prevalence of syphilis among MSM in other countries is as follows: 3.2% in Canada (2018), 2.5% in United Kingdom (2010), and 3.8%. in Spain (2013), 8.1% in Germany (2016), and 9.1% in Italy (2010) [18]. Studies performed in China (2011) and in the United States of America (2015) reported prevalence rates of 13.5% and 0.3%, respectively [6,19]. In a global meta-analysis carried out by Tsuboi et al., the global prevalence of syphilis in MSM between 2010 and 2020 was estimated to be 7.5% (_95%_CI 7.0–8.0%). The authors highlight that Latin America and the Caribbean are the regions with the highest prevalence overall (10.6% _95%_CI 8.5–12.9%) [20].

Assessing the syphilis prevalence and associated factors are important points because syphilis, similar to other STIs, disproportionately affects the MSM population. In our study, 15.2% of individuals were positive for syphilis (which is higher than the previously reported estimate); however, in 2019, the WHO reported that the prevalence of syphilis among MSM was 13.6% [18]. The difference could be explained by two factors. First, data published by the WHO only include active syphilis disease, while our findings include individuals with syphilis at diverse stages: active, latent, and resolved. Secondly, it is estimated that almost one-third of people diagnosed with syphilis have already been treated, so the reported WHO prevalence could be an underestimate [21].

Multiple variables are related to the prevalence of syphilis/HIV coinfection, and some of them are modifiable. Based on the “region”, it is possible to make political and sociocultural associations. The central region comprises a more metropolitan area with the highest social openness, while the western region is composed of a sector that is much more conservative. Furthermore, our data show that a high proportion of MSM from the West had no sexual partner in the last month; in comparison, in the center, most individuals reported having three or more sexual partners in the same period. Regarding nonmodifiable variables, age was related to exposure time. The older the individual is, the greater the probability of (1) acquiring a syphilis infection and (2) presenting antibodies against *T. pallidum,* despite being treated. Similarly, these effects can be observed for the variable “age at sexual debut with men”. Regarding “occupation”, being a student or being a student who also works could be a protective factor against syphilis infection. Furthermore, infection is correlated with younger ages.

Living with HIV/AIDS increases the risk of transmission and acquisition of other STIs, including syphilis. Additionally, associated aspects of sexual behavior in MSM who are HIV-positive, such as serosorting and seroposition, also increase the risk of syphilis [1,3,4]. In this respect, serosorting, i.e., having sexual intercourse only with partners with the same HIV status, increases the likelihood of acquiring another STI and presenting a syphilis/HIV coinfection [1,3,4]. Regarding seroposition, MSM who are exclusively receptive (referring to the sexual role) have a higher risk of becoming infected with other STIs than those who are in the intersexual and insertive sexual roles. The biological sex of sexual partners, which is considered a nonmodifiable variable, is associated with a high risk of STI acquisition in those who only have sex with men in comparison with MSM who have sexual intercourse with men and women, because anal sex is riskier (especially among those who are receptive) [1,3,21].

Being tested for HIV in the past year was neither a risk factor nor a protective factor, but rather a proxy variable that provided information about risk behavior or exposure [8]. For example, if one person experienced a risky sexual encounter, it is more likely that they took an HIV test after the event. In our study population, among MSM who were tested for HIV in the last year, a large proportion had three or more sexual partners in the last month compared with those who did not have a partner. The same effect was observed in MSM who reported practicing rewarded sex. In our work, we observed that MSM who did not practice rewarded sex presented a higher prevalence of syphilis/HIV coinfection. There are two situations regarding this point. (1) In studies such as the current one that include a questionnaire about sexual behaviors, the answers of the individuals are subject to response bias (in this context, this is also known as social desirability bias). The individuals tended to answer in a socially acceptable way [22,23,24]. Therefore, it is not surprising that most individuals reported that they do not engage in rewarded sex. (2) Reverse causality may be at play, as people who know and accept that they practice risky behavior tend to protect themselves more [25,26]. These concepts could explain the high percentage of syphilis/HIV infection among people who do not engage in rewarded sex.

A remarkable characteristic of our study population is the places where they were recruited, and MSM meeting places were recommended by the same participating MSM. These refer to clubs, movies, shopping malls, saunas, gay bars, public bathrooms, dark rooms of clubs, and others; places that have been well documented as sites used to have sexual intercourse [27,28,29]. These findings indicate that a large proportion of MSM who attend saunas or public bathrooms also practiced rewarded sex. We are not able to assess variables concerning people known to engage in sexual intercourse via the internet or social networks, which is very common in actuality, probably due to the COVID-19 pandemic.

There are few data regarding the global prevalence of syphilis/HIV coinfection. Chow et al. reported a 2.7% prevalence of coinfection in China, which is consistent with our findings (5.7%) [6]. Juárez-Figueroa et al. carried out a study to evaluate the prevalence of *T. pallidum* infection markers in PLWH receiving highly active antiretroviral therapy (HAART) in the Mexico City HIV/AIDS program at Specialized Clinic Condesa Iztapalapa (CECI). Even though they did not have national or regional representativeness, they reported an active or resolved syphilis prevalence of 44.2%, which was higher than our national result (5.7%) but close to our finding in Mexico City (39.4%) [21]. However, there is a lack of similar information in other countries. It is true that the existence of studies develops in specific places such as hospitals, clinics, cities, or regions, but the shortage of data on national prevalence complicates the comparison with our results. These findings indicate the need for performance studies about syphilis/HIV coinfection prevalence at the national level to develop and implement health policies to improve the epidemic situation of this vulnerable population around the world.

As a proposed approach to analysis, we compared the results of this work in a population of MSM who attend meeting places with public data regarding syphilis among MSM from DGE in Mexico. One limitation of the study that should be considered is that the DGE data were obtained using different methods. In this regard, comparisons between the survey (2013) (based on treponemal tests) and DGE (2013) (based on non-treponemal tests) show an increase in regional prevalence found in the first one, despite the putative overestimation of syphilis-positive cases in DGE due to the production of unspecific reagin antibodies [30]. The higher positivity in the 2013 survey could be due to MSM recruitment in specific places, which is likely to be associated with an increased likelihood of engaging in risky sexual behaviors and therefore acquiring certain STIs (including syphilis) [27,28,29]. Nonetheless, the comparison of DGE data between 2013 and 2019 aimed to highlight that even in a similar MSM population, with the very same records by state, it is easy to notice the increase in regional syphilis prevalence. The center region was unique, as there was a decrease between 2013 and 2019 (DGE data); further information is needed to explain the cause of this decrease. By way of speculation, we could point out that the health system (including epidemiological surveillance) in Mexico was saturated at the beginning of the COVID-19 pandemic. The data of DGE regarding syphilis cases from 2020 to date could be affected. So, we do not use these to hypothesize if the decrease in syphilis prevalence in the center is constant or only specific to 2019. Unfortunately, there was no MSM survey at meeting points in 2019, so a comparison with the findings presented here is not possible. So, one main limitation of this work is the source of data used to perform the comparisons. As mentioned before, the data of DGE are obtained with nontreponemal tests and clinically, and the methods vary between states. The MSM survey recruited individuals in meeting places at which this population frequently congregates, used a treponemal test to detect positive cases of syphilis, and also included an instrument validated (questionnaire) before conducting the national survey. Even with the different sources of data, the comparisons performed at an ecological scale were performed in the same type of population (MSM), in the same country (Mexico), and in two specific years (2013 and 2019). The results represent an effort to compare the syphilis prevalence by region with the data obtained yearly but with the limitations mentioned (DGE) and a survey with a validated methodology. We only hypothesize the possible scenarios of syphilis on MSM in a country with limited resources to generate specific surveys in this type of population constantly; but this highlights the need to improve and homogenize the detection of syphilis cases, especially in vulnerable populations. Considering our findings, we hypothesize that there will be an increase in syphilis prevalence for all regions due to re-emergence of the infection and because it seems that MSM are the core group for *T. pallidum* transmission [20,31,32].

## 5. Conclusions

The findings highlight the synergy between syphilis and HIV infections. Coinfection was more common in people who had an HIV test in the last year, which could indicate a perception of risk. The national prevalence of syphilis in 2013 estimated in this work was higher than the one reported in a recent meta-analysis that included data from Latin America. Even though the survey described here was performed ten years ago, the results provide a precedent to evidence the relevance to generate constant surveys in MSM. There is a need to address the risk factors described in this study and others to prevent syphilis infection, HIV infection, and syphilis/HIV coinfection. Because of the diverse clinical presentations and evolution of syphilis among PLWH, tests for these infections must be made accessible and should be promoted, especially for MSM. Likewise, it is recommended that MSM and PLWH be conscientious about serosorting and seroposition, since these sexual practices could increase the risk of acquiring other STIs, especially for those who do not favor condom use. This was also true for those attending meeting places, having multiple sexual partners, and engaging in rewarded sex because these factors also increase the risk of being infected with syphilis, HIV, or being coinfected.

In the data of the DGE and MSM survey, there is a remarkable difference in the prevalence of syphilis, HIV, and its coinfection between geographic regions within the same country. The hypothesized sceneries obtained with the comparisons of syphilis prevalence between the survey and DGE data highlight the importance of improving the detection of syphilis cases in vulnerable populations. Similar to other countries, Mexico needs to assess elements regarding syphilis/HIV coinfection and implement preventive measures among MSM.

## Figures and Tables

**Figure 1 pathogens-12-00356-f001:**
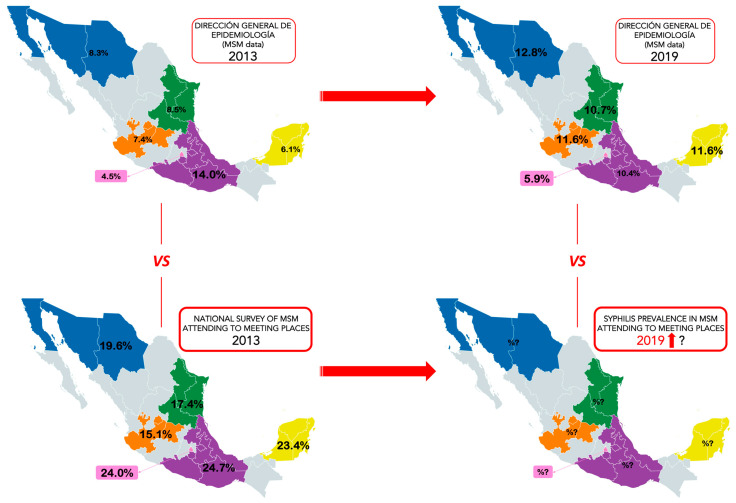
Comparison of regional prevalence of syphilis among MSM. The maps show the regional prevalence of syphilis among MSM in Mexico in different years. Top left: prevalence estimates with data on syphilis only in MSM in 2013, obtained from DGE. Bottom left: the prevalence was calculated with the results of the MSM survey (2013). Top right: prevalence by region, estimates with MSM data of DGE (2019). Lower right: hypothetical scenario of syphilis prevalence for 2019. Regions are in solid colors: Northwest (blue), Northeast (green), West (orange), Mexico City (pink), Center (purple) and South (yellow). States in gray were not considered in the analyses. The prevalence for each region is in bold. Available online: https://www.mapchart.net/mexico.html (accessed on 30 October 2022).

**Table 1 pathogens-12-00356-t001:** Descriptive of sociodemographic, clinical, and sexual behavior variables in the study population.

Variables	Categories	n	%
Region	Mexico City	1389	27.9
	Center	961	19.3
	West	894	18.0
	Northwest	693	13.9
	Northeast	741	14.9
	South	299	6.0
Aged	≥35 years	869	17.5
	30–34 years	655	13.2
	25–29 years	1126	22.6
	20–24 years	1728	34.7
	18–19 years	599	12.0
Goods	0–3	579	11.6
	4–6	2329	46.8
	7–8	2069	41.6
Occupation	Employee	3059	61.5
	Unemployed	307	6.2
	Own business	262	5.3
	Others	56	1.1
	Student	1293	26.0
Inhalant drugs (last 12 months)	Any time	235	4.7
	Never	4742	95.3
HIV infection	Positive	789	15.9
	Negative	4188	84.1
HIV test (last 12 months)	Yes	2387	48.0
	No	2590	52.0
Sexual intercourse (last 12 months)	Only men	4381	88.0
	Men and women	596	12.0
Rewarded sex	Yes	1330	26.7
	No	3647	73.3
Male sexual partners (last month)	Not answer	332	6.7
	≥6 partners	294	5.9
	2–5 partners	1269	25.5
	1 partner	2184	43.9
	Neither	898	18.0
Aged at sexual debut with men	Missing	222	4.5
	8–10 years	187	3.8
	11–15 years	1417	28.5
	16–20 years	2364	47.5
	21–25 years	603	12.1
	≥26 years	184	3.7

The population description of MSM who attend meeting points (n = 4977) and the variables considered in the analyses with categories. n: number of individuals, %: percentage.

**Table 2 pathogens-12-00356-t002:** Syphilis and associated variables of MSM from Mexico.

Variables	Categories	Syphilis (%)	OR (_95%_CI)	ORa (_95%_CI)
Region	Mexico City	24.0	**1.8 (1.4–2.2)**	**1.6 (1.3–1.9)**
	Center	24.7	**1.8 (1.5–2.3)**	**1.9 (1.5–2.5)**
	Northwest	19.6	**1.4 (1.1–1.8)**	1.2 (0.9–1.6)
	Northeast	17.4	1.2 (0.9–1.5)	1.1 (0.8–1.4)
	South	23.4	**1.7 (1.2–2.4)**	**1.6 (1.1–2.2)**
	West	15.1	1.0	1.0
Aged	≥35 years	28.2	**3.0 (2.2–4.0)**	**2.5 (1.8–3.5)**
	30–34 years	24.3	**2.4 (1.8–3.3)**	**1.9 (1.4–2.7)**
	25–29 years	23.6	**2.3 (1.8–3.1)**	**2.0 (1.5–2.8)**
	20–24 years	17.4	**1.6 (1.2–2.1)**	**1.4 (1.1–1.9)**
	18–19 years	11.7	1.0	1.0
Goods	0–3	25.6	**1.4 (1.2–1.8)**	**1.3 (1.1–1.7)**
	4–6	21.2	1.1 (0.9–1.3)	1.1 (0.9–1.3)
	7–8	19.3	1.0	1.0
Occupation	Employee	22.8	**1.7 (1.4–2.0)**	**1.4 (1.1–1.7)**
	Unemployed	19.5	**1.4 (1.0–1.9)**	1.0 (0.7–1.5)
	Own business	28.2	**2.2 (1.6–3.1)**	**1.6 (1.2–2.3)**
	Others	32.1	**2.7 (1.5–4.8)**	**1.9 (1.0–3.4)**
	Student	14.9	1.0	1.0
Inhalant drugs (last 12 months)	Any time	31.1	**1.8 (1.3–2.3)**	**1.5 (1.1–2.1)**
	Never	20.4	1.0	1.0
HIV infection	Positive	36.0	**2.5 (2.2–3.0)**	**2.0 (1.7–2.4)**
	Negative	18.1	1.0	1.0
HIV test (last 12 months)	Yes	23.9	**1.4 (1.2–1.6)**	**1.3 (1.1–1.5)**
	No	18.2	1.0	1.0
Sexual intercourse (last 12 months)	Only men	21.6	**1.5 (1.2–1.9)**	**1.4 (1.1–1.8)**
	Men and women	15.8	1.0	1.0
Rewarded sex	Yes	25.6	**1.4 (1.2–1.7)**	**1.3 (1.1–1.5)**
	No	19.2	1.0	1.0
Male sexual partners (last month)	Not answer	19.9	1.1 (0.8–1.5)	1.1 (0.8–1.5)
	≥6 partners	28.9	**1.8 (1.4–2.5)**	**1.4 (1.0–2.0)**
	2–5 partners	23.9	**1.4 (1.1–1.8)**	1.2 (0.9–1.5)
	1 partner	19.4	1.1 (0.9–1.3)	1.0 (0.8–1.3)
	Neither	18.2	1.0	1.0
Aged at sexual debut with men	Missing	15.8	0.8 (0.5–1.3)	1.1 (0.7–2.0)
	8–10 years	33.7	**2.2 (1.3–3.5)**	**2.0 (1.2–3.3)**
	11–15 years	22.7	1.2 (0.8–1.8)	1.4(0.9–2.1)
	16–20 years	20.1	1.1 (0.7–1.6)	1.3 (0.9–2.0)
	21–25 years	18.7	1.0 (0.6–1.5)	1.1 (0.7–1.6)
	≥26 years	19.0	1.0	1.0

It is possible to observe the distribution of syphilis prevalence between the different variables and the categories within them. To highlight the variables or factors associated with syphilis prevalence, the last column shows the adjusted OR (odds ratio) obtained by logistic regression for the categories with confidence intervals at 95% (_95%_CI) and underlines the categories at high risk. Because the prevalence of syphilis was >10%, the ORs could overestimate the association. Nonetheless, in bold we highlight the strongest associations.

**Table 3 pathogens-12-00356-t003:** HIV, syphilis, and coinfection among MSM from Mexico.

STATUS		Syphilis (-) HIV (-)	Syphilis (+) HIV (-)	Syphilis (-) HIV (+)	Syphilis (+) HIV (+)	*p* Value
n (%)		*3431 (68.9)*	*757 (15.2)*	*505 (10.2)*	*284 (5.7)*	
**Variables**	**Categories**					
Region	Mexico City	**25.9**	**29.3**	**33.1**	**39.4**	<0.001
	Center	18.3	22.6	19.0	23.2	
	Northwest	**14.3**	**13.3**	**13.3**	**12.3**	
	Northeast	15.8	13.5	13.7	9.5	
	South	5.5	6.9	7.7	6.3	
	West	20.2	14.4	13.3	9.2	
Aged	≥35 years	**14.9**	**21.5**	**22.2**	**28.9**	<0.001
	30–34 years	**12.0**	**13.5**	**16.8**	**20.1**	
	25–29 years	21.7	24.7	22.8	27.8	
	20–24 years	37.0	32.5	30.9	19.4	
	18–19 years	14.3	7.8	7.3	3.9	
Goods	0–3	10.7	14.0	12.7	**14.8**	0.017
	4–6	46.4	48.5	48.1	44.7	
	7–8	42.9	37.5	39.2	40.5	
Occupation	Employed	59.8	67.1	61.4	**66.2**	<0.001
	Unemployed	5.7	5.7	10.1	6.0	
	Own business	4.7	6.3	5.1	9.2	
	Others	0.8	2.0	1.8	1.1	
	Student	28.9	18.9	21.6	**7.6**	
Inhalable drugs (last 12 months)	Yes	3.8	6.7	6.3	**7.7**	<0.001
	No	96.2	93.3	93.7	92.3	
HIV test (last 12 months)	Yes	45.4	53.0	51.7	**59.5**	<0.001
	No	54.6	47.0	48.3	40.5	
Sexual intercourse (last 12 months)	Only men	86.6	89.6	91.7	**94.7**	<0.001
	Men and women	13.4	10.4	8.3	5.3	
Rewarded sex	Yes	24.2	30.3	31.9	39.1	<0.001
	No	75.8	69.7	68.1	**60.9**	
Male sexual partners (last month)	Not answer	6.6	5.8	7.9	7.7	<0.001
	≥6 partners	5.2	8.2	6.3	8.1	
	2–5 partners	24.1	28.1	27.7	31.7	
	One partner	45.1	42.5	41.8	**35.9**	
	Neither	19.0	15.3	16.2	16.5	
Aged at sexual debut with men	Missing	5.0	3.0	3.2	4.2	<0.001
	8–10 years	**2.9**	5.2	5.1	**8.5**	
	11–15 years	**27.3**	29.6	31.7	**34.2**	
	16–20 years	48.2	47.0	46.7	41.5	
	21–25 years	12.7	11.2	10.7	9.9	
	≥26	4.0	4.0	2.6	1.8	

The table shows a comparison of variables according to syphilis and HIV test results. These statuses are shown in each column (to the right) with double negatives and coinfection. In italics are the number of individuals of each status and the percentage that they represent of the total sample of MSM included in the analyses. In our study population, it is remarkable that the overall syphilis prevalence was higher than the HIV prevalence (15.2 vs. 10.2). We highlight in bold the most interesting results. All variables were statistically significant (*p* value < 0.05).

## Data Availability

The data presented in this study are available on request from the corresponding author. The data are not publicly available by confidential issue.

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
