# Peer review of "High Prevalence of Syphilis and Syphilis/HIV Coinfection among Men Who Have Sex with Men Who Attend Meeting Places in Mexico"

_pathogens, 2023, doi:10.3390/pathogens12030356_

Round 1
Reviewer 1 Report
Vergara-Ortega et al. report on syphilis/HIV-coinfections in Mexican MSM from the public health perspective. The data are interesting and worth presenting, nevertheless, I have a few recommendations.
1.) Abstract, line 20: The authors define the term “meeting places” relatively late in the discussion. For readers who are non-familiar with this nomenclature, an earlier definition (including the abstract) would be desirable.
2.) Abstract, line 27: The term “goods” (as a proxy for material wealth) is not defined in the abstract and quite superficially defined in the results chapter. An earlier and better definition would be desirable to make the interpretation of the respective results easier.
3.) Introduction, line 37: It is unclear to me why the authors consider syphilis as “re-emerging”. Has it ever stopped being prevalent in Mexico, justifying such a point of view?
4.) Materials and methods chapter, subheading “laboratory test”: Data on the diagnostic accuracy of the applied assays would make it easier for the readers to interpret the test results.
5.) Results, line 104-106: As already stated in my second point, the definition of “goods” could still be improved to make this parameter better understandable.
6.) As far as described in the paragraph on statistical assessments, no correction for multiple testing (e.g., according to Bonferroni-Holmes) was conducted. The authors should at least comment on this limitation to avoid overinterpretations of their results.
7.) Results, line 186-187) The interesting finding that “rewarded sex” was associated with less syphilis-/HIV-coinfections deserves further interpretation in the discussion. Do the authors assume a higher risk awareness in this population?
8.) Discussion, first paragraph: This paragraph is highly repetitive and could be shortened without losing relevant contents.
9.) Discussion: The authors report on very young individuals (aged 8-10) who already have experience with MSM sex and – associated – show quite increased infection rates. Is there any association with violence regarding the STI transmission in this minority or is respective information not available from the questionnaires? The authors might want to comment on this likely question of their readers in the discussion.
10.) Discussion: If such data can be made available, associations of HIV pre-exposure prophylaxis use and increased syphilis infections would be an interesting point in the discussion.
11.) Although I am not a native English speaker myself, some of the terms in the article read non-idiomatic. Language proof-reading either by the authors or by the journal seems advisable.
Author Response
Thanks for your comments.

Reviewer 2 Report
Authors state that they aimed to determine HIV/syphilis coinfection in a sample of MSM attending to meeting places.
Overall, the study is very confusing as authors mix the results obtained with specimens collected from MSM attending to meeting places with data compilation of syphilis reported in Mexico for all MSM. It is quite hard to understand how//what solid basis exist to make such a mixture. The basis for this work is from 2013, so it is also difficult to understand why authors decide to perform this work about ten years later. But then they compare with 2019 data?... Not comprehensible.
In addition, authors refer to prior publications (REFS 10, 11, 17)
Authors used a treponemic method to syphilis diagnose so one cannot understand what syphilis status of each participant. Some older men might be positive for more than 30 years…
Authors use the term “goods” but they never explain what this means.
In page 6 authors create a definition of ‘syphilis prevalence’ and ‘HIV prevalence’ that is not appropriate, the term ‘frequency’ would be more adequate.
Results are too long, discussion is too long, and conclusions are generalities.
Author Response
1) Authors state that they aimed to determine HIV/syphilis coinfection in a sample of MSM
attending to meeting places.
Yes, we aimed to determine the prevalence of syphilis/HIV coinfection in a national sample of MSM
who attend meeting places in Mexico, to evaluate factors associated with syphilis, and to compare
the syphilis prevalence of the survey and DGE data (the objective is completely described in lines 20-
24 of the abstract).
2) Overall, the study is very confusing as authors mix the results obtained with specimens
collected from MSM attending to meeting places with data compilation of syphilis reported
in Mexico for all MSM. It is quite hard to understand how//what solid basis exist to make
such a mixture. The basis for this work is from 2013, so it is also difficult to understand why
authors decide to perform this work about ten years later. But then they compare with 2019
data?... Not comprehensible. In addition, authors refer to prior publications (REFS 10, 11, 17).
We are grateful for your observation. The national survey of MSM attending meeting places was
performed in 2013 and has national and regional representativeness. Unfortunately, this survey is
the last focused only on MSM, with national representativeness and with a large study population
(approximately 5,000 MSM), so the biological samples and the information obtained is such a
valuable resource for investigation (considering resource, logistic and economical limitations to
perform a new survey). It is true that ten years later, we are working with these data, but in light of
the re-emergence of syphilis (as various authors have suggested and evidenced), we believe that
having this panorama and information in a vulnerable population such as MSM will allow us to have
a precedent for such a situation in our country. In this regard, the analysis of DGE data from 2019 in
comparison with DGE data from 2013 (comparison 1) gave us a basis to show the changes in syphilis
prevalence through the years (all regions except one increased). The comparison between DGE data
2013 and the survey of MSM 2013 (comparison 2) was performed to demonstrate the increase in the
prevalence of syphilis (thus indicating that MSM who attend meeting places could be at a high risk).
Finally, since we do not have a data source for MSM who attend meeting places for 2019, we only try
to hypothesize (based on the aforementioned increases for the other comparisons) that we would find
a scenario with a higher prevalence of syphilis for the different regions of Mexico. In addition, we
referred to other publications about the survey that do not focus on syphilis or syphilis/HIV
coinfection but on the survey itself. We apologize for the lack of clarity regarding the main idea of
the manuscript.
3) Authors used a treponemic method to syphilis diagnose so one cannot understand what
syphilis status of each participant. Some older men might be positive for more than 30
years…
Thank you. It is true that we cannot determine the syphilis stage in each participant because dried
blood spot samples were used. However, the main interest of our work was to detect all the syphilispositive
cases rather than classify them. In this regard, we used the SYPHILIS Biokit 3.0 with a
reported specificity of 100% (95%CI 98.5-100%) and sensitivity of 97.4% (95%CI 92.5-99.5%); even at
different disease stages, the kit detected 90.90% untreated primary syphilis cases and 100% treated
cases, 100% treated and untreated secondary cases, and 100% early and late latent cases (treated and
untreated) in a specific panel of reactive specimens.
4) Authors use the term “goods”, but they never explain what this means.
This is a very accurate observation, thank you very much. We added the next definition: of
participants (i.e., a categorized number of material possessions such as a car, washing machine, dryer,
or television, etc.) were concentrated in the medium and high categories., in lines 116-118 and in the
abstract (lines 31-32, i.e., a minimal number of material possessions such as a car or dryer, etc., which
served as a proxy for low economic income level).
5) In page 6 authors create a definition of ‘syphilis prevalence’ and ‘HIV prevalence’ that is not
appropriate, the term ‘frequency’ would be more adequate.
We appreciate this suggestion. However, the data presented in Table 3 (“Page 6”) are precisely the
prevalence of syphilis (15.2%) and HIV (10.2%). Both of these percentages were calculated as (number
of positives with the infection of interest/all the study population) *100, and the condition to count in
each column was the negative result for the other infection. Therefore, we respectfully decided to
maintain the term prevalence.
6) Results are too long, discussion is too long, and conclusions are generalities.
We are grateful for your particular observation. The manuscript includes the results by subheadings
to facilitate their presentation. Although there are no subheadings in the discussion, we tried to
present a point of view for each set of results. We are sorry if these parts are too long; maybe this is
because each subheading is accompanied with a table. In contrast, the conclusions are generalities
with the idea to present a wide picture of the manuscript without repetition of results or discussion
points.

Reviewer 3 Report
I appreciate the opportunity to critically review this interesting manuscript in which the authors aimed to determine the prevalence of syphilis/HIV in a sample of MSM in Mexico. The manuscript is well written and the objective was achieved. I have only minor comments to make.
1. In the statistical analysis: It is possible to infer that logistic regression models were used to estimate ORs and CIs, but this must be explicit in the manuscript. Related to the above, the prevalence of the event of interest was high (>10%), so I suggest discussing (briefly) the impact of the above on the estimated ORs.
2. Information from the DGE regarding the prevalence of syphilis in the population of interest was used. It is highly probable that the source of these data is non treponemal tests so, if so, I suggest including it in the limitations of the study.
3. Men who made their sexual debut with other men at a very young age (8 - 10 years) had the highest chance of being identified as syphilis cases. This seems particularly worrying to me since these are cases that are potentially related to a history of sexual assault. I suggest also briefly discussing the above.
Author Response
Thanks for your comments.

Round 2
Reviewer 2 Report
Authors addressed some of the problems found in the first version of the paper.
Discussion now includes two new sentences that should disappear: lines 357-360. Authors here state that the use of non-treponemal tests could overestimate DGE findings. This is useless when authors found much higher rates, so I do not see the interest of adding these sentences. It even highlights a basic wrong principle of this paper that is to compare, do statistics and take conclusions based on non-comparable population groups: the DGE and the sampling of 4977-selected MSM.
Moreover, authors may have detected higher rates because they used a treponemal test, which means that they are counting “all persons who have now or have had syphilis in the past”; DGE data, based on non-treponemal tests, might rely in more recent syphilis cases, as non-treponemal antibodies tend to disappear with treatment or x years after primary syphilis.
Therefore, authors should/must discuss the hypothetical impact of using different methodologies for counting the number of syphilis cases, considering that they don’t count exactly the same persons.
Authors continue with a bizarre point of view stating that DGE data between 2013 and 2019 include “the same MSM population”?? The same?? People are surely different between 2013 and 2019!
Also, authors state that the health system in Mexico was saturated in 2019 because of COVID19?? How can that be? The first COVID19 cases in Mexico, as for many western countries, were described in the end of February 2020…
The first paragraph of conclusions has no relation with results so it cannot be included in conclusions.
Author Response
Thanks for the comments. The changes in the manuscript are highlight in aqua.

Round 3
Reviewer 2 Report
Authors answered to the problems raised by their paper.
Suggestions:
The term STDs is used twice and the authors discriminate the mean of the abbreviation. But later in the text authors use STI and STIs instead, without discriminating the mean of the abbreviation. I guess could use only STD or only STI, but if they prefer to use both, they discriminate the mean of the abbreviation at the first time it is used.
I suggest the rewriting of the sentences where this study and DGE results are discussed based on possible impact of the methodology:
“Overall, comparisons between the survey (2013) and DGE (2013) show a rise in regional prevalences found in the first one. This could be due to MSM recruitment in specific places, which is likely to be associated with an increased likelihood of engaging in risky sexual behaviours and therefore acquiring some STIs (including syphilis) (27–29). Additionally, one limitation of the study to consider is that the DGE data were obtained by different methods, including nontreponemal tests (30). Therefore, the prevalence of syphilis-positive cases could be overestimated by other infection cases due to the production of reagin antibodies. Nonetheless, the comparison of DGE data between 2013 and 2019 aimed to highlight that even in the same MSM population, with the very same records by state, it is easy to notice the increase in regional syphilis prevalence.”
To:
“One limitation of the study to consider is that the DGE data were obtained by different methods. In this regard, comparisons between the survey (2013) (based on treponemal tests) and DGE (2013) (based on non-treponemal tests), show a rise in regional prevalences found in the first one, despite the putative overestimation of syphilis-positive cases in DGE due to the production of unspecific reagin antibodies (30). The higher positivity in the 2013 survey could be due to MSM recruitment in specific places, which is likely to be associated with an increased likelihood of engaging in risky sexual behaviours and therefore acquiring some STIs (including syphilis) (27–29). Nonetheless, the comparison of DGE data between 2013 and 2019 aimed to highlight that even in a similar MSM population, with the very same records by state, it is easy to notice the increase in regional syphilis prevalence.”
Author Response
Thanks for the suggestions. All the changes are highlight in fuchsia.
